# Camelpox Virus in Western Kazakhstan: Assessment of the Role of Local Fauna as Reservoirs of Infection

**DOI:** 10.3390/v16101626

**Published:** 2024-10-17

**Authors:** Yerbol Bulatov, Sholpan Turyskeldy, Ruslan Abitayev, Abdurakhman Usembai, Zhanna Sametova, Zhanat Kondybayeva, Alina Kurmasheva, Dana Mazbayeva, Asselya Kyrgyzbayeva, Kamshat Shorayeva, Zhanat Amanova, Dariya Toktyrova

**Affiliations:** Research Institute for Biological Safety Problems, Gvardeiskiy 080409, Kazakhstan; ye.bulatov@biosafety.kz (Y.B.); sh.smankizi@biosafety.kz (S.T.); r.abitaev@biosafety.kz (R.A.); a.ussenbay@biosafety.kz (A.U.); zh.sametova@biosafety.kz (Z.S.); zh.kondybaeva@biosafety.kz (Z.K.); a.kurmasheva@biosafety.kz (A.K.); mazbayevad@mail.ru (D.M.); asselya91@icloud.com (A.K.); k.shorayeva@biosafety.kz (K.S.); zh.amanova@biosafety.kz (Z.A.)

**Keywords:** camelpox virus, *Orthopoxvirus*, camel, reservoirs, ticks, blood-sucking insects, rodents

## Abstract

This article investigates the role of local fauna in Western Kazakhstan as potential reservoirs of the camelpox virus (CMLV). The study emphasizes analyzing possible sources and transmission pathways of the virus using polymerase chain reaction (PCR) and serological methods, including virus neutralization tests and enzyme-linked immunosorbent assays (ELISA). Samples were collected from both young and adult camels, as well as rodents, ticks and blood-sucking insects in the Mangystau and Atyrau regions. The PCR results revealed the absence of viral DNA in rodents, ticks and blood-sucking insects; also, the ELISA test did not detect specific antibodies in rodents. These findings suggest that these groups of fauna likely do not play a significant role in the maintenance and spread of CMLV. Consequently, the primary sources of transmission are likely other factors, potentially including the camels themselves. The study’s results indicate the need to reassess current hypotheses regarding infection reservoirs and to explore alternative sources to enhance strategies for the control and prevention of the camelpox virus.

## 1. Introduction

Camel breeding is one of the traditional branches of agriculture in the Republic of Kazakhstan and is widespread in many other countries, particularly in regions with hot and dry climates. The Republic of Kazakhstan encompasses various natural areas, including dry steppe and semi-desert areas, which can be developed through camel breeding. Such regions include the Mangystau, Atyrau, Aktobe, Kyzylorda, Zhambyl, Almaty and East Kazakhstan regions. In recent years, the increase in the number of farms engaged in camel breeding indicates Kazakhstan’s ambition to take a leading position in the international arena.

Bacterial and viral infectious diseases are the primary constraints on livestock growth and farm development [1]. One of these most common diseases is camelpox [2].

The camelpox virus (CMLV) belongs to the genus *Orthopoxvirus*, which is part of the extensive family *Poxviridae*. Phylogenetic analysis of *Camelpoxvirus* shows that it is closely related to the *Variola virus* (VARV), the causative agent of human smallpox, which has been eradicated worldwide [3]. Although camelpox is a zoonotic disease that can negatively impact agriculture, its significance for public health is currently low, as the disease is rare in humans and typically causes only mild symptoms [4].

In Kazakhstan and neighboring countries such as Uzbekistan, Turkmenistan and Mongolia, double-humped camels (*Camelus bactrianus*) are a possible major source of camelpox virus. Camels living in these regions often interact closely with each other, which can contribute to the spread of the virus [5]. Transmission of camelpox occurs through direct contact with infected animals, either via skin abrasions or aerosols. Materials such as scabs, saliva and secretions from affected camels can release the virus into the environment, including water sources, which then become infection reservoirs. Various studies have shown that the incidence of camelpox outbreaks increases during rainy seasons, often leading to more severe cases. This may be attributed to enhanced virus stability in moist conditions, facilitating further transmission to susceptible animals. Additionally, the presence of arthropods during these seasons, which may act as mechanical vectors for the virus, could play a role, such as ticks and mosquitoes. Among tick species, *Hyalomma dromedarii* has been identified as the predominant species affecting camels, accounting for ~90% of the total. Wernery et al. [6] supported this idea through their isolation of CMLV from *Hyalomma dromedarii* ticks [7,8,9].

In Kazakhstan, outbreaks of camelpox are recorded with a certain frequency, especially in the southern and western regions, where there is a high density of camel populations. A similar situation is observed in Uzbekistan and Turkmenistan, where camels also play an important role in agriculture [10]. Mongolia, with its significant number of camels, also faces periodic outbreaks of the disease, especially in areas with unfavorable climatic conditions that can contribute to the spread of infection [11].

Similar outbreaks were systematically observed on the territory of the country in the Mangystau and Atyrau regions in 1930, 1942–1943, 1965–1967, 1968–1969, 1996 and 2020 [12]. During the outbreak in 1996, the epidemic spread widely in three districts of the Mangystau region; 830 cases and 43 deaths out of 8 thousand camels were registered. Subsequently, the viral strain that caused the epidemic situation was isolated, sequenced and deposited in the GenBank database (No. AF438165.1) [13]. The last outbreak of camelpox in the Mangystau region in 2019–2020 was confirmed in 70 camels in the Karakiya and Beineu districts. In addition, an outbreak of camelpox was recorded in the Mangystau and Munaily districts. The evaluation of statistical data on the incidence of camelpox in these regions of the Republic of Kazakhstan allowed us to determine the cycle of occurrence of epizootics, which is about 10–20 years [2,14].

This research analyzes possible reservoirs of camelpox virus in the western part of Kazakhstan, as well as studies on the likelihood of the disease. In addition, information on the epidemiology and methods of transmission of camelpox to possible reservoirs was also included.

## 2. Materials and Methods

### 2.1. Sampling

In 2023, spring and autumn expeditions were conducted to collect biological samples from economic entities of the Mangystau region: Beineu, Karakiya, Mangystau, Munaily and Tupkaragan. In 2024, similar work was carried out in the Atyrau region in the following districts: Zhylyoi, Inder, Isatay, Kyzylkoga, Kurmangazy, Makat and Makhambetbet. The expedition areas are indicated on the map shown in Figure 1.

To conduct experiments, the necessary materials from animals were obtained with the permission of the Veterinary Control and Supervision Committee of the Ministry of Agriculture of the Republic of Kazakhstan.

During the field expeditions, biological samples were collected: blood and serum from camels and rodents, as well as ticks and blood-sucking insects. The process of collecting biological samples is illustrated in Figure 2.

Camels: The Mangystau and Atyrau regions of Kazakhstan are home to various species of camels, such as the double-humped camel (*Camelus bactrianus*) and the single-humped camel (*Camelus dromedarius*) [15], which play a significant role in the lives of local residents.

Blood sampling from camels was performed using a 21 G × 1.5-inch needle inserted into the jugular vein in the upper third, with samples collected in vacuum tubes containing EDTA K2, each with a volume of 5 mL. For serum samples, special vacuum tubes with a coagulation activator and gel were used.

Ticks: Ticks play an important role in ecosystems and can be carriers of various diseases. Table 1 shows the species of ticks that are adapted to the local climatic conditions of the Mangystau and Atyrau regions [16,17].

In the Atyrau and Mangystau regions of Kazakhstan, species of ticks from the genus *Hyalomma*, which parasitize camels, are found. These regions have suitable climatic conditions that facilitate the spread of these ticks. The tick activity season in the Atyrau and Mangystau regions begins in spring (April–May) and continues through to autumn (September–October). During the summer months, tick activity may decrease, but they remain a threat to animals throughout the period [18].

The lifespan of *Hyalomma* ticks that parasitize camels can vary depending on environmental conditions and the tick’s developmental stage. Table 2 provides the approximate lifespan of these ticks. It is important to note that ticks go through several developmental stages: egg, larva, nymph and adult tick. Each stage has its own lifespan characteristics.

Temperature and humidity strongly affect the lifespan of these ticks. In hot and dry conditions, life expectancy may decrease. Ticks need the host’s blood at every stage of their development, with the exception of the egg stage. The adult male does not need a blood meal, whereas the adult female requires one to lay eggs. Without a suitable host, ticks can fall into a dormant state and remain alive for several months to a year.

The collection of ticks was carried out from the surface of the camels and in the rooms where the camels were kept using personal protective equipment. The collected ectoparasites were placed in 100 mL plastic jars filled with 70% alcohol.

Blood-sucking insects: In addition to ticks, the Atyrau and Mangystau regions are home to various species of blood-sucking insects (Table 3), which can also play an important role in the transmission of diseases among animals and humans. These insects actively feed on the blood of their hosts, especially in the warmer seasons.

To catch blood-sucking insects, special light traps with ultraviolet lamps were installed in places where camels were kept. Insects collected in traps were placed in 200 mL plastic jars containing 100 mL of 70% alcohol. Collected blood-sucking arthropods after their capture should be sorted by gender and parity (determined visually under a dissecting magnifying glass) by wing pattern and size [23]. Before determining the species, blood-sucking insects are treated with tobacco smoke, in order to immobilize the insect, and then packed in 20 copies, in individual packaging.

Rodents: The Mangystau and Atyrau regions of Kazakhstan are distinguished by their semi-desert and desert landscapes, which affects the species composition of rodents living in these regions. Table 4 below shows the approximate number of some rodent species found in these areas:

Rodents were caught alive using live traps in the form of nets or boxes, in which 1 cm^3^ pieces of bread soaked in vegetable oil were placed. After collecting the necessary amounts of blood for research, the rodents were released.

Biological samples were randomly selected and immediately delivered to the laboratory in a Dewar vessel with liquid nitrogen. Upon arrival at the laboratory, all samples were stored at −40 °C until the start of research work.

Based on the weather forecast from the KazHydroMet meteorological service [28], as well as the population density of the regions and the total number of camels, the following average number of samples was collected in each area (Figure 3):

### 2.2. Preparation of Materials

For the experiment, age-related differences and the immune status of camels were studied. Blood and serum samples obtained from camels and rodents were aliquoted, labeled and stored at −20 °C until further analysis, such as DNA and virus extraction.

Morphological analysis and identification of collected ticks and blood-feeding insects were conducted with adherence to biological safety measures, using a stereomicroscope (Leitz Diavert, Wetzlar, Germany). The research plans were approved by the Bioethics Supervisory Board of the Scientific Research Institute for Biosafety Problems with permission No. 1511/011 prior to the start of the research. Throughout the study, institutional codes, operational procedures and recommendations for the treatment of animals were strictly followed.

### 2.3. Detection of CMLV-Specific Antibodies

For serological research aimed at detecting antibodies against camelpox, the Camelpox Virus Antibody (CMLV-Ab) ELISA Kit (Abbexa Ltd., Cambridge, UK) was used, following the instructions provided with the commercial kit. Additionally, camelpox virus neutralization tests were performed.

The virus neutralization test (VNT) was conducted in accordance with the WOAH guidelines for terrestrial use (2012) [29]. This method uses blood serum, which was diluted in a ratio from 1:2 to 1:128 and mixed in equal volumes with the vaccine strain KM-40 at a concentration of 100 TCID_50/mL_ [30].

### 2.4. DNA Extraction

Total genomic DNA was extracted from blood samples of camels, rodents and from blood-feeding insects and ticks using the Blood and Tissue DNeasy Kit (Qiagen, Hilden, Germany) according to the provided protocol. Prior to DNA extraction, the ticks and blood-feeding insects were frozen and then homogenized. The DNA was eluted in 100 µL of elution buffer, free from DNA/RNA contaminants, and stored at −20 °C until further analysis.

### 2.5. Molecular Genetic Analysis

To diagnose camelpox virus, the PCR-RV diagnostic system (QuantStudio 5) was used with a set of Primerdesign LTD reagents (Genesig, Eastleigh, UK). Briefly, the volume of 1 reaction was 20 µL, of which 15 µL was the reaction mixture and 5 µL was the test DNA. The condition for PCR amplification consisted of 2 steps: Step 1—shutter speed 95 °C for 2 min; Step 2—95 °C for 10 s, 60 °C for 60 s with FAM signal detection. A total of 50 cycles were performed. Samples that were above the threshold and showed up positive before the 32nd cycle were considered positive.

### 2.6. Statistical Analysis

Statistical data processing was carried out using EpiInfo 7.2.2.2 (CDC) software and GraphPad Prism 8.0.1. The creation of maps and cartographic designations were performed in the Esri ArcGIS 2.2 program.

### 2.7. Bioethics

All manipulations related to animals were carried out in accordance with the Law on Responsible Treatment of Animals of the Republic of Kazakhstan (Law No. 97-VII LRK, Republic of Kazakhstan, 30 December 2021) and other applicable recommendations. The research plans were approved by the Bioethics Supervisory Board of the Scientific Research Institute for Biosafety Problems with permission No. 1511/011 prior to the start of the research. Throughout the study, institutional codes, operational procedures and recommendations for the treatment of animals were strictly followed.

## 3. Results

From March to April and from September to October 2023, as well as from May to June 2024, a field expedition was conducted to the western regions of Kazakhstan, where camel breeding is widespread. The study was conducted against the background of an outbreak of camelpox registered in 2019 in the Beineu, Karakiya, Mangystau and Munaily districts of the Mangystau region. The main purpose of the expedition was to identify potential reservoirs of camelpox virus in the fauna of Mangystau and Atyrau. During the expedition, samples were collected from camels of various age groups, as well as from rodents, blood-sucking insects and ticks. The age differences among the camels from which samples were collected for the study are shown in Table 5.

In 20 farms of the Mangystau region, from which the materials for the study were obtained, all camels older than 2 years were vaccinated against camelpox with a live attenuated vaccine (strain KM-40), as well as against other epidemic diseases such as rabies, plague and anthrax. In the Atyrau region, camels from which blood samples were taken, according to documents, were vaccinated against camelpox in 2023. The collection of materials in 2024 coincided with the period before the annual revaccination. In total, biomaterials were collected from 720 camels.

During the analysis of the collected blood-sucking insects, a thorough examination of their various parts, including the veins of the wings, heads and abdomens, was carried out. As a result of the study, it was possible to identify insects as *Anopheles hyrcanus* and *Culex pipiens* belonging to the family *Culicidae* spp. (Table 3). These species have significant epidemiological significance, as they can be carriers of various infectious diseases.

Morphological analysis of the ticks revealed that the detected ticks belong to the ticks found in the Atyrau and Mangystau regions indicated in Table 1 above. The *Hyalomma dromedarii* species, which is unique to camels, was selected from the collected ticks. This type of tick plays a key role in the transmission of certain diseases and is an important object of study for assessing the risks associated with infectious diseases of camels.

### 3.1. Molecular Diagnostic Results

The results of the PCR analysis for detecting the causative agent of camelpox showed positive results in 19 (6.3%) out of 300 samples from camels in the Mangystau region and in 22 (5.24%) out of 420 samples from the Atyrau region. In total, 41 positive samples were identified, which constitutes 6% of all examined samples.

The analysis revealed that in the Mangystau region, the proportion of positive samples among young animals not vaccinated against camelpox was 12 (63.15%) of the total positive samples. In the Atyrau region, this proportion was 13 (59.1%). Positive samples for the presence of the camelpox virus are shown in Table 6. These data indicate that young animals are particularly susceptible to the camelpox virus and are at a higher risk of infection.

Other objects, from which DNA samples isolated from ticks (*n* = 205) and mosquitoes (*n* = 587) were used for analysis, were tested for the presence of camelpox virus using the pool testing method. The results of testing of all samples are shown in Figure 4. It was found that only one sample showed an insignificant signal at 43 cycles of PCR-RT, which indicates the possible presence of the virus in this sample but does not confirm its presence in significant quantities. The camelpox virus was not detected in 27 blood samples of the small gopher, which confirms the absence of this species as a reservoir of the virus in the studied region.

In general, the course and outcome of camelpox can vary based on age, sex and circulating strains of CMLV, which differ in virulence. Thus, risk factors associated with higher morbidity from camelpox include the average age of animals (under 4 years), the rainy season, the introduction of new camels into the herd and shared water sources [31].

### 3.2. Results of Serological Tests

Serological analyses showed that of the 150 studied serums of adult camels in the Mangystau region, 86 (57.3%) have antibodies against the camelpox virus, and of 210 adult camels in the Atyrau region, antibodies were found only in 23 (15.3%) camels. All serum samples from young camels in both areas showed negative results. In addition, the blood serum of the rodent *Spermophilus pygmaeus n* = 23 and *Meriones meridianus n* = 24 also did not give positive results (Table 7).

In the Mangystau region, a higher number of antibodies and viral particles were recorded compared to the Atyrau region, which may indicate a higher activity of the virus in this region. This difference may be related to epidemiological characteristics, conditions of keeping camels and the level of vaccination in both areas. The absence of antibodies in all the studied young camels in both areas indicates that this age group may not have been exposed to the virus as well as vaccinated.

## 4. Discussion

Many viruses of the *Orthomyxoviridae* family are able to infect and reproduce in a wide range of host species and taxa [32]. Currently, little is known about the mechanisms of transmission and conservation of zoonotic OPXV in natural conditions. These viruses often remain mysterious and manifest themselves only in secondary infections in species that are probably not the main natural hosts of the virus [33,34,35,36,37]. Thus, most of the knowledge about OPXV is obtained as a result of accidental interspecific transmission, usually from an unknown source, including humans, predators or domestic animals [31,38,39,40].

Camelpox is registered in almost all countries where camels are bred [8]. In the period from 2017 to 2021, outbreaks of camelpox were recorded in 12 countries [2]. An analysis of cases over the past eight years has shown that sporadic cases have been reported in four countries (Israel, Iraq, Eritrea and Kazakhstan), while in the remaining eight countries (Iran, Libya, Oman, Palestine, Saudi Arabia, Somalia, Tunisia and Ethiopia), the disease is endemic [2,41].

Environmental and geographical conditions in Kazakhstan and neighboring countries play an important role in the spread of the camelpox virus. In areas with a dry climate and limited access to water, camels are often kept in close groups, which increase the risk of infection. In addition, seasonal camel migrations between regions can contribute to the spread of the virus over long distances [42]. Increased attention to the control of animal trafficking, especially from areas with known outbreaks, as well as the provision of strict quarantine measures and medical examinations, can significantly reduce the risk of spreading the virus.

Significant outbreaks of camelpox in Kazakhstan have been recorded since 1930. The Mangystau region, which borders Turkmenistan and Uzbekistan, previously suffered from epizootics in Turkmenistan when this country was part of the USSR. In the period from 1965 to 1969, large epizootics in Turkmenistan spread to the territory of the Kazakh SSR, causing epizootics in the Mangystau region [42]. In 2018 and 2019, cases of camelpox were observed in the Balkan province of Turkmenistan, adjacent to the Mangystau region, which led to the death of both young and adult animals [43]. Turkmenistan also borders Iran, where outbreaks of this infection are recorded annually [44]. Sporadic cases and even epizootics of camelpox may occur in Uzbekistan, but there is currently no specific information about such events.

According to media reports, in the spring months of 2019, a massive outbreak of camelpox occurred in the western regions of Turkmenistan, as a result of which many camels died. Local residents reported an unknown disease that periodically affects camels and cows, leading to the death of livestock. However, the authorities denied information about the disease that occurs among camels, claiming that camels die of starvation [45,46]. During a field expedition conducted in 2023–2024, our researchers found out that camels from Turkmenistan were imported to Kazakhstan in 2019–2020. After their transportation, the disease began to spread among local camels located near transport routes. In addition, in 2019, a massive outbreak of camelpox occurred in the Beine district of the Mangystau region, which was confirmed by laboratory tests.

The camelpox outbreak in Turkmenistan and the Mangystau region of Kazakhstan in 2019 illustrates how rapidly the disease can spread in the absence of adequate control measures. The cross-border nature of the regions necessitates coordinated efforts for monitoring and control. The intensification of trade activities between Kazakhstan and Turkmenistan, particularly without proper quarantine controls, can facilitate the spread of the virus. Strengthening cooperation between countries is essential to prevent the spread of the disease.

Currently, besides preventive measures, the only method of protection against camelpox involves vaccines based on CMLV. However, these vaccines are not widely used and currently protect only camel calves older than 6 months.

Protection against CMLV (*camelpox virus*) relies on both humoral and cellular immune responses in animals. Vaccination against camelpox can be achieved with either live attenuated or inactivated vaccines. Although the immunity provided by the attenuated vaccine is long-lasting, revaccination is recommended every 6–9 months. In contrast, inactivated vaccines typically require annual revaccination [47].

Following the successful completion of field trials in Kazakhstan in 2023, a vaccination program was launched using a domestic live attenuated camelpox vaccine. According to the results of the field trials, the humoral immunity in vaccinated camels diminishes after 6 months post-vaccination [14].

Based on the serological analyses conducted during the study, antibodies against the camelpox virus were detected in 57.33% of the samples from the Mangystau region, while in the Atyrau region, this figure was only 10.95%. It is important to note that camels in the Atyrau region were not vaccinated in 2024. These results suggest that the seroprevalence of the camelpox virus varies with age, as vaccination is only administered to camels older than six months. Consequently, PCR analysis for detecting the Camelpox pathogen revealed positive results in 6% of adult camels and 61% of juvenile camels among all tested samples. The presence of the virus in asymptomatic camels confirms their role as hidden reservoirs, complicating disease control and highlighting the need for regular monitoring and preventive measures.

Analysis of the literature on CMLV reveals that there is information suggesting the potential involvement of wild rodents [48,49,50], arthropods and insects [6,51] in the transmission of camelpox among camels. However, these data are presented mainly as hypotheses and have not been thoroughly investigated. It is noteworthy that the CMLV shares similarities with the CPXV pathogen, for which domestic and wild rodents are known primary reservoirs [48]. J. Chantrey et al. [49], demonstrated that the primary reservoirs for bovinepox are field voles. They also hypothesize that bank voles might not sustain the infection alone, which could explain the absence of bovinepox in Ireland, where voles are not present. Infection among wild rodents varies seasonally, and these fluctuations likely account for the pronounced seasonal disease occurrence in incidental hosts such as humans and domestic cats [49,50].

In our studies, almost all samples from rodents (*Meriones meridianus* and *Spermophilus pygmaeus*) and blood-feeding insects yielded negative results, suggesting their minimal role in the epidemiological chain of camelpox transmission. However, these results do not completely rule out their potential as reservoirs, considering that several rodent and blood-feeding insect species in Western Kazakhstan remain unstudied. This highlights the need for further investigation into other potential reservoirs and transmission pathways, including other rodents and domestic animal species, to ensure a comprehensive approach to disease control. Additionally, the results underscore the importance of monitoring insect populations, as they might play a significant role in camelpox transmission. Seasonal and geographical variations in viral activity among ticks and insects should be considered when developing control strategies.

The primary reservoir for camelpox remains the camels themselves. However, the role of other animals (such as rodents of the species *Allactaga major*, *Meriones tamariscinus* and *Apodemus sylvaticus*), fauna (blood-feeding insects like *Culicidae*, *Simuliidae*, *Ceratopogonidae* and *Haematopinus tuberculatus*; other tick species of the genus *Hyalomma* spp.) and environmental factors also requires further investigation, as some of the studied materials proved insufficient. For example, during the collection of biological samples in the Beineu district, disinfection against ticks and insects was conducted. During the visit to Atyrau, blood-feeding insects could not be found in two districts due to unfavorable weather conditions. Additionally, in some areas, it was impossible to collect rodents for biological samples due to adverse regional conditions and poor weather, such as flooding in Atyrau region in 2024. Efforts are ongoing to isolate the camelpox virus from positive samples and to gather additional field data for more thorough analysis.

Overall, monitoring natural populations over time can more precisely establish the dynamics of circulating pathogens. The obtained information is valuable for understanding host–pathogen interactions and assessing the risk of zoonotic diseases. Studying camelpox virus reservoirs in Kazakhstan and neighboring countries is crucial for developing effective disease prevention and control strategies.

## Figures and Tables

**Figure 1 viruses-16-01626-f001:**
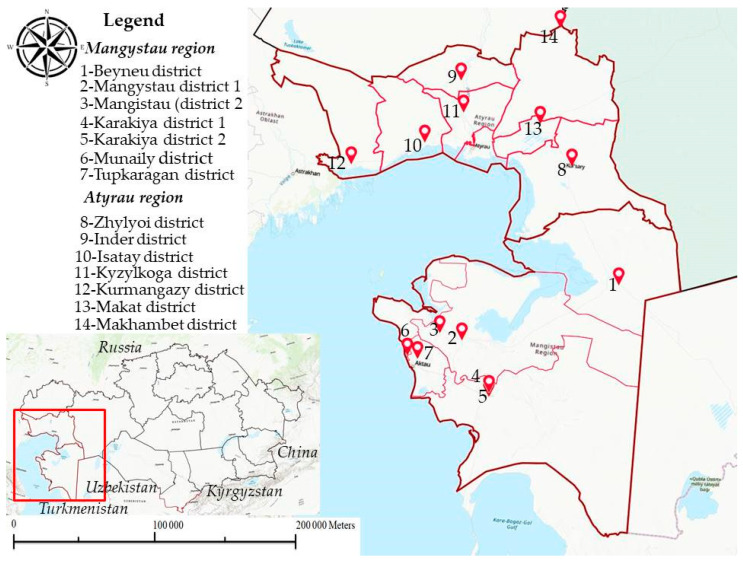
Map of Mangystau and Atyrau regions and geographical location of biological samples.

**Figure 2 viruses-16-01626-f002:**
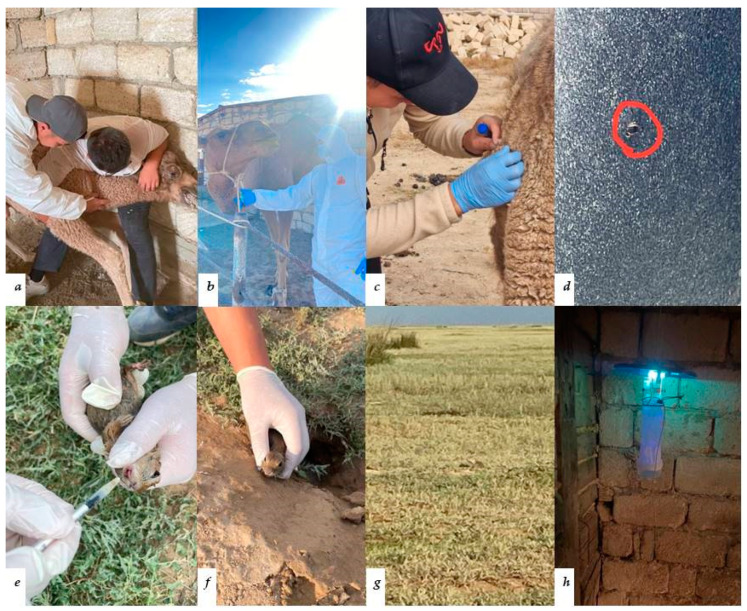
Collection of biological samples from the Mangystau and Atyrau regions: (**a**) blood collection from a camel calf; (**b**) blood collection from a camel; (**c**) tick collection from the surface of a camel; (**d**) tick on the camel’s skin; (**e**) blood collection from a small ground squirrel; (**f**) rodent nest; (**g**) remote observation of a small ground squirrel; (**h**) UV light trap for blood-feeding insects.

**Figure 3 viruses-16-01626-f003:**
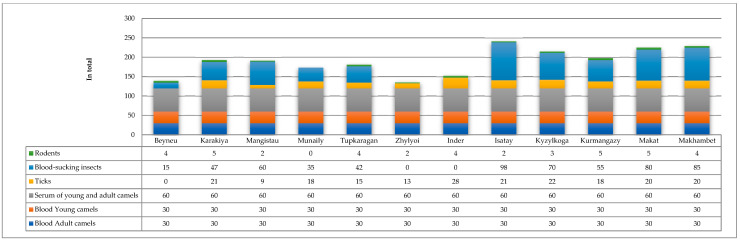
Biological sample collection data for 2023–2024 from the Mangystau and Atyrau regions of Kazakhstan.

**Figure 4 viruses-16-01626-f004:**
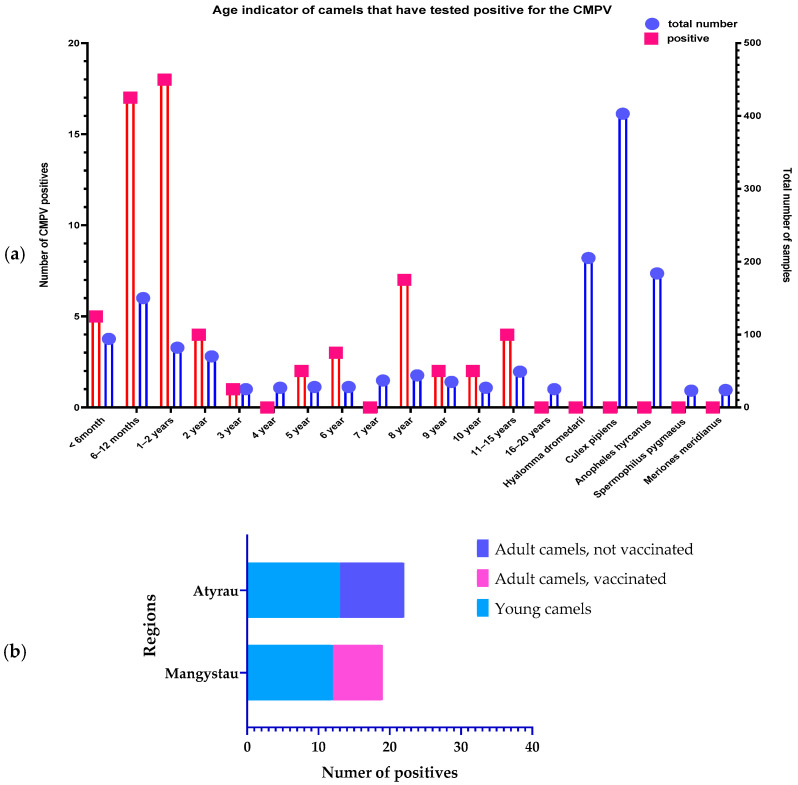
(**a**) Quantitative size of samples with positive CMLV in the PCR-RT test system depending on the age of the camels and the species; (**b**) comparison of data between regions with different levels of immunization.

**Table 1 viruses-16-01626-t001:** Species composition of ticks in Mangystau and Atyrau regions.

Species of Tick	Regions of Western Kazakhstan	
	Mangystau	Atyrau
*Dermacentor marginatus*	+	+
*Dermacentor pictus*	+	
*Hyalomma asiaticum*	+	+
*Hyalomma scupense*	+	+
*Hyalomma anatolicum*		+
*Hyalomma numidiana*	+	
*Hyalomma marginatum*		+
*Hyalomma dromedarii*	+	+
*Ixodes laguri*		+
*Rhipicephalus rossicus*		+
*Ornithodoros tartacovskii*	+	
*Rhipicephalu sschulzei*		+
*Rhipicephalus turanicus*		+
*Dermacentor niveus*		+
*Rhipicephalus pumilio*		+
*Ixodes occultus*	+	+
*Haemaphysalis erinacei*		+

**Table 2 viruses-16-01626-t002:** The lifespan of camel ticks.

Types of Ticks	Life Cycle					Reference
	Eggs	Larvae	Nymphs	Adults	Length of Life	
*Hyalomma dromedarii*—the main parasite of camels in arid regions, including the Atyrau and Mangystau regions, is the Hyalomma tick. This tick is widely distributed in the semi-desert and desert areas of Kazakhstan.	A few weeks	1–2 months	3–4 months	6–8 months, under favorable conditions up to 1 year	From several months to more than a year, depending on environmental conditions	[19]
*Hyalomma marginatum*—is widespread in the Atyrau and Mangystau regions and parasitizes large mammals, including camels.	2–4 weeks	1–3 months	2–4 months	6–10 months, sometimes up to 1 year	On average, from 1 to 2 years	[16]
*Hyalomma anatolicum*—is mainly found in the southern regions of Kazakhstan. It is a vector of various pathogens.	2–3 weeks	1–2 months	2–3 months	8–10 months, up to 1 year under favorable conditions	Up to 1–1.5 years, depending on the conditions	[17]

**Table 3 viruses-16-01626-t003:** Species of blood-sucking insects found in Atyrau and Mangystau regions.

Species of Blood-Sucking Insect	№	Species	Mangystau Region	Atyrau Region	Season of Activity	Reference
Mosquitoes (*Culicidae*) are important carriers of various viruses and parasites, including malaria, West Nile fever and other diseases. Their numbers depend on the availability of water, as they breed in stagnant reservoirs.	1	*Anopheles maculipennis*	+++	+++	Spring (April–May) to autumn (September–October)	[20]
	2	*Anopheles hyrcanus*	+++	++++		
	3	*Anopheles atroparvus*		+		
	4	*Uranotaenia unguiculata*		++		
	5	*Mansonia richiardii*		++		
	6	*Aedes mariae*		+		
	7	*Aedes vexans*	++++	+++		
	8	*Aedes cinereus*		+++		
	9	*Aedes intrudens*	+	+		
	10	*Ochlerotatus caspius*	++++	++++		
	11	*Ochlerotatus dorsalis*	+++	++++		
	12	*Ochlerotatus behningi*		+		
	13	*Ochlerotatus cyprius*		+		
	14	*Ochlerotatus flavescens*		++		
	15	*Ochlerotatus excrucians*		++++		
	16	*Ochlerotatus subdiversus*	++	+		
	17	*Ochlerotatus detritus*	+	++		
	18	*Ochlerotatus cataphylla*		+		
	19	*Ochlerotatus leucomelas*		+		
	20	*Ochlerotatus communis*		++		
	21	*Ochlerotatus cantans*	++	++		
	22	*Culex modestus*	++++	+++		
	23	*Culex pusillus*	++	++		
	24	*Culex pipiens*	+++	+++		
Black flies (*Simuliidae*) can carry various pathogens.			Under-researched		Spring (April–June) and summer (July–August)	[21]
Biting midges (*Ceratopogonidae*) can carry viruses that cause diseases in animals and humans. They are active mainly at dusk and at night.			Underexplored		Summer (June–August)	[22]

Note: ++++—mass; +++—a significant amount; ++—a small amount; +—rare.

**Table 4 viruses-16-01626-t004:** Species composition of rodents living in Atyrau and Mangystau regions.

Rodent Species	Mangystau Region (Approximate Number)	Atyrau Region (Approximate Number)	Reference
Midday gerbil (*Meriones meridianus*)	1000–2000	1200–2500	[24]
Little ground squirrel (*Spermophilus pygmaeus*)	500–1000	400–800	[25]
Great jerboa (*Allactaga major*)	500–1000	600–1200	[25]
Tamarisk jird (*Meriones tamariscinus*)	800–1500	900–1600	[26]
Wood mouse (*Apodemus sylvaticus*)	600–1200	700–1300	[27]

Note: The data are approximate and based on studies on rodent fauna in Kazakhstan.

**Table 5 viruses-16-01626-t005:** Ratios of camel numbers by age group.

Age of Camels	Mangystau, *n* = 300	Atyrau, *n* = 420	Percentage Ratio, *n* = 720, 100%
Up to 6 months	20	75	13.19
6–12 months	99	51	20.83
1–1.5 years	31	45	10.55
1.5–2 years	20	56	10.55
3 years	10	15	3.47
4 years	13	18	4.31
5 years	14	15	4.03
6 years	16	14	4.16
7 years	24	13	5.14
8 years	14	30	6.11
9 years	11	24	4.86
10 years	11	7	2.51
11–15 years	7	42	6.81
16–20 years	9	15	3.34
Over 20 years	1	-	0.14

**Table 6 viruses-16-01626-t006:** List of positive samples from Mangystau and Atyrau regions and age differences of camels.

Mangystau Region			Atyrau Region		
Beineu District			Zhylyoi District		
№	Name of the Sample	Age of the Camel	№	Name of the Sample	Age of the Camel
1	2B	6 m.	1	11T	6 y. 5 m.
2	27B	6 m.	2	22T	8 y. 3 m.
3	8T	6 y	3	40B	10 m.
4	14T	8 y	4	42B	10 m.
5	19T	6 y	5	59B	10 m.
6	23T	2 y	Inder district		
Mangystau district			6	73T	13 y. 4 m.
7	42B	1 y. 6 m.	7	83T	8 y. 5 m.
8	46B	6 m.	8	99B	1 y. 9 m.
9	47B	7 m.	9	101B	1 y. 8 m.
Karakiya district			10	106B	1 y. 7 m.
10	61B	6 m.	11	107B	1 y. 8 m.
11	63B	7 m.	12	113B	1 y. 9 m.
12	85B	4 m.	13	120B	2 y. 7 m.
13	89B	1 y. 6 m.	Isatay district—all samples are negative		
14	72T	2 y. 10 m.	Kyzylkoga district		
Munaily district			14	188T	5 y. 2 m.
15	104B	7 m.	15	200T	9 y. 4 m.
16	106B	6 m.	16	202T	8 y. 4 m.
Tupkaragan district			17	233B	9 m.
17	129B	6 m.	Kurmangazy district		
18	142T	3 y. 6 m.	18	256T	10 y. 4 m.
19	144T	6 y. 6 m.	19	292B	1 y. 5 m.
			Makat district		
			20	347B	5 m.
			Makhambet district		
			21	402B	4 m.
			22	403B	4 m.

Note. The table shows the age of camels according to animal documents. T—camels; B—young camels.

**Table 7 viruses-16-01626-t007:** Results of serological tests.

Type of Animal	Mangystau Region			Atyrau Region		
	Total	VNT	ELISA	Total	VNT	ELISA
Young camels	150	-	-	212	-	-
Adult camels	150	86	4	208	23	21
*Spermophilus pygmaeus*	10	-	-	13	-	-
*Meriones meridianus*	10	-	-	14	-	-
	320			447		
Total number of antibody positives		86			23	

Note: (-)—negative results; VNT—virus neutralization test.

## Data Availability

Data are contained within the article.

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
