# Peer review of "Camelpox Virus in Western Kazakhstan: Assessment of the Role of Local Fauna as Reservoirs of Infection"

_viruses, 2024, doi:10.3390/v16101626_

Round 1
Reviewer 1 Report
Comments and Suggestions for Authors
The authors assess the role of certain rodents and insects as potential reservoirs for camelpox. The results of the assessment of rodents, ticks and insects were that no camalpox genomes were detected in the tested samples and in addition no camelpox specific antibodies were detected in the rodents sampled. Having a negative results does not mean that these species do not play a role in camelpoxvirus transmission, but shows that these species may play a minor role in camelpox transmission.
Minor issues.
Line 19 please rewrite as antibodies were not assessed by ELISA in ticks and blood sucking insects.
Line 36 camelpox
Line 69 "change will also be included" to "was also included"
Line 152 in b the camel looks smaller then a is this also a camel calf?
Table 5 Please try to make the table with the regions able to fit in the table without splitting the word or in the case of long words more than once.
Line 169 as a space after the .
Line 188 remove "the"
Line 189 was used with a set of Primerdesign LTD reagents (Genesig, Great Britain).
Line 227 with a live "attenuated" vaccine
Line 251 3.2. "Molecular diagnostic results"
Figure 5 for the PCR where a positive sample is identified if it has a CT value less than 32 fir instance in Mangystau Beyneu I only see 1 sample which has a CT value of less than 32, yet there are 6 samples listed. This is an issue with most of of the results in figure 5. Can you please confirm the results for the real-time PCR as correct or modify as required. Once done please modify the paper with the proper positive samples.
Also please check the spelling of Mangystau in Figure 5
Line 293 "may not have been exposed to the virus as well as vaccinated."
Comments on the Quality of English Language
See comment to authors.
Author Response
Response to Reviewer 1 Comments
Thank you very much for taking the time to review this manuscript. Your feedback has allowed us to see it from a new perspective and provided recommendations for improving our work. We are grateful for your support in making our research more reliable and ethically sound. We look forward to your suggestions in our revised manuscript. Once again, thank you for your valuable insights. We greatly appreciate your professionalism and the time you dedicated to the review process. Please find detailed responses to your comments below, along with the corresponding changes highlighted in the tracked changes section of the resubmitted files. |
Does the introduction provide sufficient background and include all relevant references? Yes/Can be improved/Must be improved/Not applicable Are all the cited references relevant to the research? Yes/Can be improved/Must be improved/Not applicable Is the research design appropriate? Yes/Can be improved/Must be improved/Not applicable Are the methods adequately described? Yes/Can be improved/Must be improved/Not applicable Are the results clearly presented? Yes/Can be improved/Must be improved/Not applicable Are the conclusions supported by the results? Yes/Can be improved/Must be improved/Not applicable |
|
Comments 1: Line 19 please rewrite as antibodies were not assessed by ELISA in ticks and blood sucking insects. Response 1: Thank you for your thoughtful review and recommendation. You are absolutely correct. We were inattentive. We agree with this comment. The sentence “The PCR and ELISA results revealed an absence of viral DNA and specific antibodies in rodents, ticks, and blood-sucking insects.” has been rewritten as follows: “The PCR results revealed the absence of viral DNA in rodents, ticks, and blood-sucking insects; additionally, the ELISA test did not detect specific antibodies in rodents.” 18-20 lines Comments 2: Line 36 camelpox Response 2: Thank you for pointing this out. We agree with this comment. The error has been corrected. Comments 3: Line 69 "change will also be included" to "was also included" Line 80 Response 3: Thank you very much for your thoughtful guidance. We agree with this comment. The error has been corrected. Comments 4: Line 152 in b the camel looks smaller then a is this also a camel calf? Response 4: You are absolutely right. We were inattentive and have replaced the figure accordingly. Line 156 Comments 5: Table 5 Please try to make the table with the regions able to fit in the table without splitting the word or in the case of long words more than once. Response 5: We appreciate your attention to detail. We have made adjustments to ensure that the region names fit in the table and that long words are only split once. Comments 6: Line 169 as a space after the . Response 6: Thank you for pointing this out. We agree with this comment. The errors have been corrected. Comments 7: Line 188 remove "the" Response 7: Thank you for pointing this out. We agree with this comment. The errors have been corrected. Comments 8: Line 189 was used with a set of Primerdesign LTD reagents (Genesig, Great Britain). Response 8: Thank you for pointing this out. We agree with this comment. The errors have been corrected. Comments 9: Line 227 with a live "attenuated" vaccine Response 9: Thank you for bringing this to our attention. We were inattentive. We agree with this comment. The errors have been corrected. Comments 10: Line 251 3.2. "Molecular diagnostic results" Response 10: We fully agree with your suggestion. We appreciate your sharp observation regarding the heading. We understand that clarity is important and have made the necessary changes. Your input is valuable, and we believe these changes will enhance the overall quality of the work. Comments 11: Figure 5 for the PCR where a positive sample is identified if it has a CT value less than 32 fir instance in Mangystau Beyneu I only see 1 sample which has a CT value of less than 32, yet there are 6 samples listed. This is an issue with most of of the results in figure 5. Can you please confirm the results for the real-time PCR as correct or modify as required. Once done please modify the paper with the proper positive samples. Also please check the spelling of Mangystau in Figure 5 Response 11: We are very grateful for your attention to this detail. We agree with this comment. Given the large number of samples, the PCR results were presented in parts. Considering that our figure caused visual difficulties, we have replaced the figure with a table (Table 7). Comments 12: Line 293 "may not have been exposed to the virus as well as vaccinated." Response 12: Thank you for pointing this out. We agree with this comment. The error has been corrected. |
Point 1:Minor editing of English language required Response 1: We have implemented all the changes you suggested regarding the English language and have also eliminated as many inaccurate expressions as possible found in the manuscript text. |
Reviewer 2 Report
Comments and Suggestions for Authors
Introduction
line 29 – In all countries? or where?
line 41 – damage to the farm?
line 45 – close contact to? humans?
Table 1. Type of ticks. These are not types but species. Quantity of ticks. Who measured these numbers and how? Tick abundance is not equal in various territories. Appearance of ticks is patchy. Here you find a lot, 10 meters away nothing. These „quantity” data should be avoided or give details how these numbers were measured. And at what territories? Dry grasslands of west Kazakstan, or everywhere? In Europe, in hilly regions „quantity” of tick species are the same?
References Table 1. References (15 from the 52) written in Russian are not forbidden to cite, but mostly unavailable or not understandable for most part of the scientific audience (e.g. ref 12, 14, 15, 16). Would be best to avoided. I tried, these references are not avaiable in ncbi/pubmed. Practically unavailable for the public. These are practically not references.
The very same for Table 3. ref 18-20.
line 116 – Do adult males of these species need blood meal?
Comments on the Quality of English Language
spelling
line 169 – USA).Inter a space is missing
lines 172, 182 and extra row is missing
line 192 95 c instead of °C
line 373-374 – CMLV or CML virus
references - (2 points after ref 14).
space mistakes ref 1.
references
Author Response
Response to Reviewer 2 Comments
Thank you very much for taking the time to review this manuscript. Your feedback has allowed us to see it from a new perspective and provided recommendations for improving our work. We are grateful for your support in making our research more reliable and ethically sound. We look forward to your suggestions in our revised manuscript. Once again, thank you for your valuable insights. We greatly appreciate your professionalism and the time you dedicated to the review process. Please find detailed responses to your comments below, along with the corresponding changes highlighted in the tracked changes section of the resubmitted files. |
Does the introduction provide sufficient background and include all relevant references? Yes/Can be improved/Must be improved/Not applicable Are all the cited references relevant to the research? Yes/Can be improved/Must be improved/Not applicable Is the research design appropriate? Yes/Can be improved/Must be improved/Not applicable Are the methods adequately described? Yes/Can be improved/Must be improved/Not applicable Are the results clearly presented? Yes/Can be improved/Must be improved/Not applicable Are the conclusions supported by the results? Yes/Can be improved/Must be improved/Not applicable |
|
Comments 1: Introduction line 29 – In all countries? or where? Response 1: You are absolutely right. We were inattentive. We agree with this comment. The sentence “Camel breeding is one of the traditional branches of agriculture.” has been rewritten as follows: “Camel breeding is one of the traditional branches of agriculture in the Republic of Kazakhstan and is widespread in many other countries, especially in regions with hot and dry climates.” Comments 2: line 41 – damage to the farm? Response 2: Thank you for pointing this out. We agree with this comment. The phrase "to the farm" has been rewritten as "to agriculture." Comments 3: line 45 – close contact to? humans? Response 3: We are very grateful for your attention to this detail. We agree with this comment. The phrase "kept in close contact" has been rewritten as "interact closely with each other." Here, we meant that the large number of camels in these areas may be another important factor in the spread of camel pox, as they will be closely associated with one another. Comments 4: Table 1. Type of ticks. These are not types but species. Quantity of ticks. Who measured these numbers and how? Tick abundance is not equal in various territories. Appearance of ticks is patchy. Here you find a lot, 10 meters away nothing. These „quantity” data should be avoided or give details how these numbers were measured. And at what territories? Dry grasslands of west Kazakstan, or everywhere? In Europe, in hilly regions „quantity” of tick species are the same?
Response 4: You are absolutely right. We appreciate your suggestion and will definitely revise and update the table. We will replace "Type of tick" with "Species of tick." We were not attentive enough and overlooked this point. We fully agree with your perspective regarding the number of ticks, as the data was collected from outdated sources and the internet.
Comments 5: References Table 1. References (15 from the 52) written in Russian are not forbidden to cite, but mostly unavailable or not understandable for most part of the scientific audience (e.g. ref 12, 14, 15, 16). Would be best to avoided. I tried, these references are not avaiable in ncbi/pubmed. Practically unavailable for the public. These are practically not references. Response 5: Thank you very much for your thoughtful recommendation. We understand your concerns regarding the Russian references that may be inaccessible or unclear to a wider scientific audience. We agree that such references can pose challenges. From the sources you mentioned, some have been improved, and we have made changes to the corresponding sections of the manuscript (the order numbers are listed below). Given the value of the materials from sources 15, 22, and 42, we would like to retain references to these sources, despite the information being published in journals not included in ranking databases. For sources 11 and 12, the information is presented in two references, and we have removed 12 since it is a scientific thesis by the co-author of 11. For sources 14, 16, 18, 19, 23, and 24, we have added references, some of which link directly to the book. Comments 6: The very same for Table 3. ref 18-20.
Response 6: Thank you for your help and valuable feedback. The table has been changed and improved according to your recommendations. We have also adjusted the references to make them more accessible.
Comments 7: line 116 – Do adult males of these species need blood meal?
Response 7: Thank you for your comment. We agree with this feedback. Ticks go through three stages in their search for a host for a blood meal: larva, nymph, and adult female. The larva must feed to become a nymph; the nymph must feed to grow into an adult; and the adult female requires a blood meal to lay eggs. The adult male does not need blood. We have added a sentence about this in the paragraph.
Comments 8: spelling line 169 – USA).Inter a space is missing lines 172, 182 and extra row is missing line 192 95 c instead of °C line 373-374 – CMLV or CML virus references - (2 points after ref 14). space mistakes ref 1. Response 8: You are absolutely right. We agree with this comment. The errors have been corrected. Some articles and old books are available only in Russian, as many researchers do not have the opportunity to publish their work in ranked journals. If you would like, we can send you some of these sources. |
Point 1: Moderate editing of English language required. Response 1: We have eliminated as many inaccurate expressions in English as possible found in the manuscript text. |
Reviewer 3 Report
Comments and Suggestions for Authors
Table 3. in common English, Simuliids are black flies. Ceratopogonids are biting midges or commonly called noseeums, not wood lice.
Figure 2 is not necessary.
Table 5. Serum from which animals?
Figure 3 would be best presented as a table with age cohorts by year at the top and the numbers of samples stratified by area of sampling.
Figure 4 is not necessary.
Figure 5 would be best shown as a bar graph or table showing numbers tested/numbers positive. The line graph implies change over time, which this is not.
The paragraph starting on line 265 is confusing. A simple statement indicating that no PCR positive samples were found in how many ticks tested and how many mosquitoes were tested by species and how many rodents were tested by species. With this simple statement, the ticks, mosquitoes and rodents can be eliminated from Figure 6, simplifying it.
The introduction makes the case for the importance of camels and their poxvirus disease. It needs to end with a clear statement of objective of the study, which appears to be the question: is there evidence of transmission of CMLV by ticks and biting insects and infection of rodents in an area of active virus transmission to and among camels.
The discussion repeats information that is in the introduction and is needless duplication. Lines 348-350 is unnecessary as it has nothing to do with the objectives. If there are important points in lines 296-380 that are not included in the introduction, they can be added there. The real discussion starts at line 351 but needs the simple statement that this study found no evidence of transmission by ticks or or mosquitoes of presence of the virus in rodents in areas where camels were infected. Then cite the evidence that supports that statement.
Comments on the Quality of English Language
The English terms need copy editing to make the text more like those commonly used English in journals. There are a few places where the author's meaning is not clear.
Author Response
Response to Reviewer 3 Comments
Thank you very much for taking the time to review this manuscript. Your feedback has allowed us to see it from a new perspective and provided recommendations for improving our work. We are grateful for your support in making our research more reliable and ethically sound. We look forward to your suggestions in our revised manuscript. Once again, thank you for your valuable insights. We greatly appreciate your professionalism and the time you dedicated to the review process. Please find detailed responses to your comments below, along with the corresponding changes highlighted in the tracked changes section of the resubmitted files. |
Does the introduction provide sufficient background and include all relevant references? Yes/Can be improved/Must be improved/Not applicable Are all the cited references relevant to the research? Yes/Can be improved/Must be improved/Not applicable Is the research design appropriate? Yes/Can be improved/Must be improved/Not applicable Are the methods adequately described? Yes/Can be improved/Must be improved/Not applicable Are the results clearly presented? Yes/Can be improved/Must be improved/Not applicable Are the conclusions supported by the results? Yes/Can be improved/Must be improved/Not applicable |
|
Comments 1: Table 3. in common English, Simuliids are black flies. Ceratopogonids are biting midges or commonly called noseeums, not wood lice. Response 1: Thank you for your clarification. We agree with this comment. We appreciate the information about Simuliids and Ceratopogonids. The errors in our work have been corrected (Midges as Black flies and Woodlice as Biting midges), and we have taken your remarks into account. Comments 2: Figure 2 is not necessary. Response 2: We appreciate your attention to detail. We aimed to present the information as clearly as possible to better highlight the key points. Comments 3: Table 5. Serum from which animals? Response 3: We are grateful for your remark. We agree with this comment. We have corrected this point from (Serum) to (serum of young and adult camels). Comments 4: Figure 3 would be best presented as a table with age cohorts by year at the top and the numbers of samples stratified by area of sampling. Response 4: Thank you for your suggestion. We agree that Figure 3 will be better presented as a table. We have replaced Figure 3 with Table 6. This will indeed help convey the information more effectively. Comments 5: Figure 4 is not necessary. Response 5: Thank you for your remark. We would like to keep the illustration (Figure 4) of ticks and mosquitoes for visualization, as it helps better illustrate the information being presented. Comments 6: Figure 5 would be best shown as a bar graph or table showing numbers tested/numbers positive. The line graph implies change over time, which this is not. Response 6: Thank you for your suggestion. We agree that Figure 5 will be clearer if presented as a table displaying the number of tested positive cases. We have replaced Figure 5 as Table 7. Comments 7: The paragraph starting on line 265 is confusing. A simple statement indicating that no PCR positive samples were found in how many ticks tested and how many mosquitoes were tested by species and how many rodents were tested by species. With this simple statement, the ticks, mosquitoes and rodents can be eliminated from Figure 6, simplifying it. Response 7: We appreciate your attention to detail. We would like to keep this paragraph, as it provides valuable information on the tick, rodent, and mosquito species for which we obtained negative results. This adds context to all the samples tested and helps better understand the study. Comments 8: The introduction makes the case for the importance of camels and their poxvirus disease. It needs to end with a clear statement of objective of the study, which appears to be the question: is there evidence of transmission of CMLV by ticks and biting insects and infection of rodents in an area of active virus transmission to and among camels. The discussion repeats information that is in the introduction and is needless duplication. Lines 348-350 is unnecessary as it has nothing to do with the objectives. If there are important points in lines 296-380 that are not included in the introduction, they can be added there. The real discussion starts at line 351 but needs the simple statement that this study found no evidence of transmission by ticks or or mosquitoes of presence of the virus in rodents in areas where camels were infected. Then cite the evidence that supports that statement. Response 8: Thank you for your comments and constructive feedback. We agree that the introduction should clearly state the study's objective regarding the transmission of CMLV by ticks and biting insects, as well as the infection of rodents. We have taken your suggestion into account and added the relevant points to the introduction. Additionally, we will review the discussion and remove lines 348-350. In the text of the article, we have included reference [51] to a source containing information about the transmission of the camelpox virus. There may also be information here that might interest you: https://doi.org/10.3390/vaccines9080912 and https://www.ncbi.nlm.nih.gov/pmc/articles/PMC3832703/. Thank you once again for your helpful comments. |
Point 1: The English terms need copy editing to make the text more like those commonly used English in journals. There are a few places where the author's meaning is not clear Response 1: We have implemented all the changes you suggested regarding the English language and have also eliminated as many inaccurate expressions as possible found in the manuscript text. |
Round 2
Reviewer 2 Report
Comments and Suggestions for Authors
I have no additional comments, what I had already made.
Comments on the Quality of English LanguageIt is OK.
Author Response
Response to Reviewer 2 Comments
We want to express our sincere gratitude for your thorough review of our manuscript. We appreciate the time and effort you dedicated to providing constructive comments and recommendations, which have significantly helped us improve the quality of our work. In response to your feedback, we made the necessary changes. Each of your comments was carefully considered, and we tried to incorporate all your recommendations from the first and second rounds. Once again, thank you for your contribution to our work. We greatly value your professionalism and the time you devoted to the review process.
|
Does the introduction provide sufficient background and include all relevant references? Yes/Can be improved/Must be improved/Not applicable Are all the cited references relevant to the research? Yes/Can be improved/Must be improved/Not applicable Is the research design appropriate? Yes/Can be improved/Must be improved/Not applicable Are the methods adequately described? Yes/Can be improved/Must be improved/Not applicable Are the results clearly presented? Yes/Can be improved/Must be improved/Not applicable Are the conclusions supported by the results? Yes/Can be improved/Must be improved/Not applicable |
|
Comments 1: I have no additional comments, what I had already made. Response 1: Thank you for your comment and for your attention to our materials. We have also reviewed your comments from the first round again and made some minor changes during the editing process.
|
Point 1: Comments on the Quality of English Language Comments 1: It is OK. Response 1: We are grateful for your assistance in enhancing our work. Your recommendations are extremely valuable to us, and we hold them in high regard. |
Reviewer 3 Report
Comments and Suggestions for Authors
The paper is much improved. The tables are easy to understand.
Line 270 -275 . Figure 43? These photographs are not necessary.
Author Response
Response to Reviewer 3 Comments
|
1. Summary We would like to express our sincere gratitude for your thorough review of our manuscript. We appreciate the time and effort you dedicated to providing constructive comments and recommendations, which have significantly improved the quality of our work. In response to your feedback, we have made the necessary revisions. Each of your comments was carefully considered, and we made every effort to incorporate your suggestions. Once again, thank you for your contribution to our work. We greatly value your professionalism and the time you devoted to the review process. |
|
2. Questions for General Evaluation Reviewer’s Evaluation Response and Revisions Does the introduction provide sufficient background and include all relevant references? Yes/Can be improved/Must be improved/Not applicable Are all the cited references relevant to the research? Yes/Can be improved/Must be improved/Not applicable Is the research design appropriate? Yes/Can be improved/Must be improved/Not applicable Are the methods adequately described? Yes/Can be improved/Must be improved/Not applicable Are the results clearly presented? Yes/Can be improved/Must be improved/Not applicable Are the conclusions supported by the results? Yes/Can be improved/Must be improved/Not applicable |
|
3. Point-by-point response to Comments and Suggestions for Authors Comments 1: Line 270 -275 . Figure 43? These photographs are not necessary. Response 1: Thank you for your comments and your attention to our materials. During the editing process, the mention of "Figure 4" was changed to "Figure 3." As a result, both the old and corrected versions are highlighted in red in the text to make the changes easier to notice. Following your recommendations, we have removed this figure.
|
Point 1: English language fine. No issues detected. Response 1: We are grateful for your assistance in enhancing our work. Your recommendations are extremely valuable to us, and we hold them in high regard. |